# Variations in the Spatial Distribution of Smart Parcel Lockers in the Central Metropolitan Region of Tianjin, China: A Comparative Analysis before and after COVID-19

Mengyue Ding [1,2], Nadeem Ullah [1] 🆔, Sara Grigoryan [1], Yike Hu [1,*] and Yan Song [2]

1   School of Architecture, Tianjin University, Tianjin 300072, China; dmengyue@unc.edu (M.D.);
    6120000154@tju.edu.cn (N.U.); 6121000169@tju.edu.cn (S.G.)
2   Department of City and Regional Planning, University of North Carolina at Chapel Hill,
    Chapel Hill, NC 27599, USA; ys@email.unc.edu
*   Correspondence: huyike11@tju.edu.cn

**Abstract:** The COVID-19 pandemic has led to a significant increase in e-commerce, which has prompted residents to shift their purchasing habits from offline to online. As a result, Smart Parcel Lockers (SPLs) have emerged as an accessible end-to-end delivery service that fits into the pandemic strategy of maintaining social distance and no-contact protocols. Although numerous studies have examined SPLs from various perspectives, few have analyzed their spatial distribution from an urban planning perspective, which could enhance the development of other disciplines in this field. To address this gap, we investigate the distribution of SPLs in Tianjin's central urban area before and after the pandemic (i.e., 2019 and 2022) using kernel density estimation, average nearest neighbor analysis, standard deviation elliptic, and geographical detector. Our results show that, in three years, the number of SPLs has increased from 51 to 479, and a majority were installed in residential communities (i.e., 92.2% in 2019, and 97.7% in 2022). We find that SPLs were distributed randomly before the pandemic, but after the pandemic, SPLs agglomerated and followed Tianjin's development pattern. We identify eight influential factors on the spatial distribution of SPLs and discuss their individual and compound effects. Our discussion highlights potential spatial distribution analysis, such as dynamic layout planning, to improve the allocation of SPLs in city planning and city logistics.

**Keywords:** smart parcel lockers; self-service technology; self-service facilities; last-delivery logistics; kernel density estimation; average nearest neighbor; standard deviational ellipse; geographical detector; Tianjin; COVID-19

## 1. Introduction

Human lifestyle and e-commerce have undergone significant changes as a result of the abrupt and long-lasting COVID-19 [1]. Many countries had policies requiring citizens to stay at home due to the risk of infection from gathering, contributing to the shift in lifestyle from offline to online [2–4]. This shift had a substantial impact on traditional face-to-face business, while e-commerce experienced rapid expansion [1,5]. Globally, e-commerce sales increased by 22.7% in 2021, 9.7% in 2022, and are expected to increase by 10.4% in 2023 [6]. According to data from Statista, the Chinese e-commerce trade increased by 14.1% in 2021, accounting for more than half of global e-commerce retail sales [7]. As such, the demand for delivery has increased significantly. China's express delivery service providers handled 108.3 billion pieces of business in 2021, a 29.92% increase from 2020 [8]. Last-mile delivery, the final and costliest stage of the supply chain, as well as the only one stage that directly engages with customers, is seen as being the most crucial [9–11].

Home delivery (HD) and collection-and-delivery points (CDPs) are the two main types of last-mile delivery [12]. In recent years, the conventional HD has imposed heavy societal costs (e.g., road congestion, noise, and air pollution) [13,14], as well as a high rate of failure

in the first-time delivery [15–17]. The need to deliver parcels a second and even a third time aggravate the traffic, noise, and pollution issues. When compared to traditional door-to-door delivery, the CDPs mode is considered beneficial from both societal and operational perspectives [13,18,19]. According to [15], consumers collecting their parcels from self-collection facilities can reduce carbon emissions by up to 83%, resulting in profound environmental improvement and significant operating cost reduction.

Compared with the two main types of CDPs, attended CDPs (ACDPs) and unattended CDPs (UCDPs), UCDPs are acknowledged to have more benefits than ACDPs. The primary distinction between these two main types of CDPs is whether or not an employee is present to assist with collection. ACDPs are inconvenient due to restrictions on off-duty hours, despite the fact that they are frequently developed within stores or shops in residential areas. UCDPs, on the contrary, are usually installed near residential and office buildings and accessible 24/7, making it convenient for customers to pick up parcels along their way home or to the office at any time. As a result, UCDPs, which typically take the form of Smart Parcel Lockers (SPLs) in last-mile delivery [20,21], are hence gaining popularity [12].

SPL, also known as the shared reception box [22], self-service parcel pickup machine [21], Modular Bentobox [23], modular pack station for parcels [24], or automated parcel station, is a self-service facility (SSF), which uses self-service technology (SST) to allow customers to produce and consume services without requiring direct assistance from company employees [25,26]. During the pandemic, some academics studied customer behavior and discovered that customers valued simplified service and self-service more [27,28]. Customers who are accustomed to online services may feel uneasy when receiving in-person service and may avoid social situations [5,27]. Under the constraints of maintaining social distance and "no contact" behavior norms, SPL is not only a convenient facility, but also an unavoidable choice for physical and mental health.

Currently, more than 20 countries around the world have deployed parcel lockers [29]. Prominent examples of SPLs are ByBox in the UK, PackStation in Germany, InPost in Poland [18], and POPStation in Singapore [30]. In China, SPLs have been included as part of urban and rural public infrastructure to be constructed [31]. Here are examples: Cainiao Station and Hive Box, two nationwide last-mile delivery service providers in China. Cainiao Station has established more than 30,000 community-based SPLs and more than 1900 college-based SPLs to date, covering more than 280 cities across China [32]; Hive Box operates more than 2 million parcels per day and provides services to more than 400 million customers in total [33]. It was predicted that SPLs would develop further and play a more prominent role in the provision of services [34,35]. Therefore, it is necessary to investigate the existing SPLs distribution and enlighten the future.

Although numerous studies have been published on SPLs, none have examined the temporal and spatial variations before and after the pandemic to demonstrate SPLs diffusion. Furthermore, to the best of the authors' knowledge, no research has been conducted on the factors that influenced changes in SPLs spatial distribution. These are the major gaps in knowledge that this study will try to fill.

The remainder of this article is organized as follows. The following section focuses on the introduction of research materials and methods. Section 3 presents the results of the spatial distribution between pre- and post-pandemic periods, as well as the influential factors. Following that, we summarized the findings and concluded with a number of implications for resilient SPL planning and management in high-density urban areas. Section 5 contains the conclusion and a few limitations.

## 2. Materials and Methods

### 2.1. Study Area

For this study, the central Tianjin region is selected. Tianjin is a city with 16 districts, a population of over 13.73 million people, and an area of 11,916.85 km$^2$ [36]. As one of China's four municipalities, Tianjin is also an e-commerce powerhouse. Tianjin's annual express delivery volumes were 1.234 billion pieces in 2021 and 0.928 billion pieces in 2020,

representing a 33.0% and 33.1% increase, respectively [37]. Meanwhile, national annual volumes of express delivery business were 108.3 billion pieces, a 29.9% increase, and 83.36 billion pieces, a 31.2% increase, respectively [8]. Tianjin's express delivery business grew at a faster rate than the national average.

The research region is in Tianjin and spans six districts between 39°2′0″ N and 39°10′0″ N and 117°8′0″ E and 117°18′0″ E (Heping District, Hedong District, Hexi District, Nankai District, and Hongqiao District). Figure 1 shows the study area with China in the backdrop, Tianjin next, and then the research area itself. It is the city's central urban area for political, cultural, and economic activities, with a land area of 576.160 km$^2$ (13.29% of Tianjin) and a population of 4.06 million people (29.26% of Tianjin, according to the People's Republic of China's seventh Census) [36]. The Haihe River flows from northwest to southeast through Tianjin's center, establishing Tianjin's urban development structure. Furthermore, the urbanization rate of the study area is 100% [36]. Due to its high population density and rapid urbanization, the central Tianjin region accounts for the majority of online shopping and express delivery services. As a result, we selected central Tianjin as a representative study area.

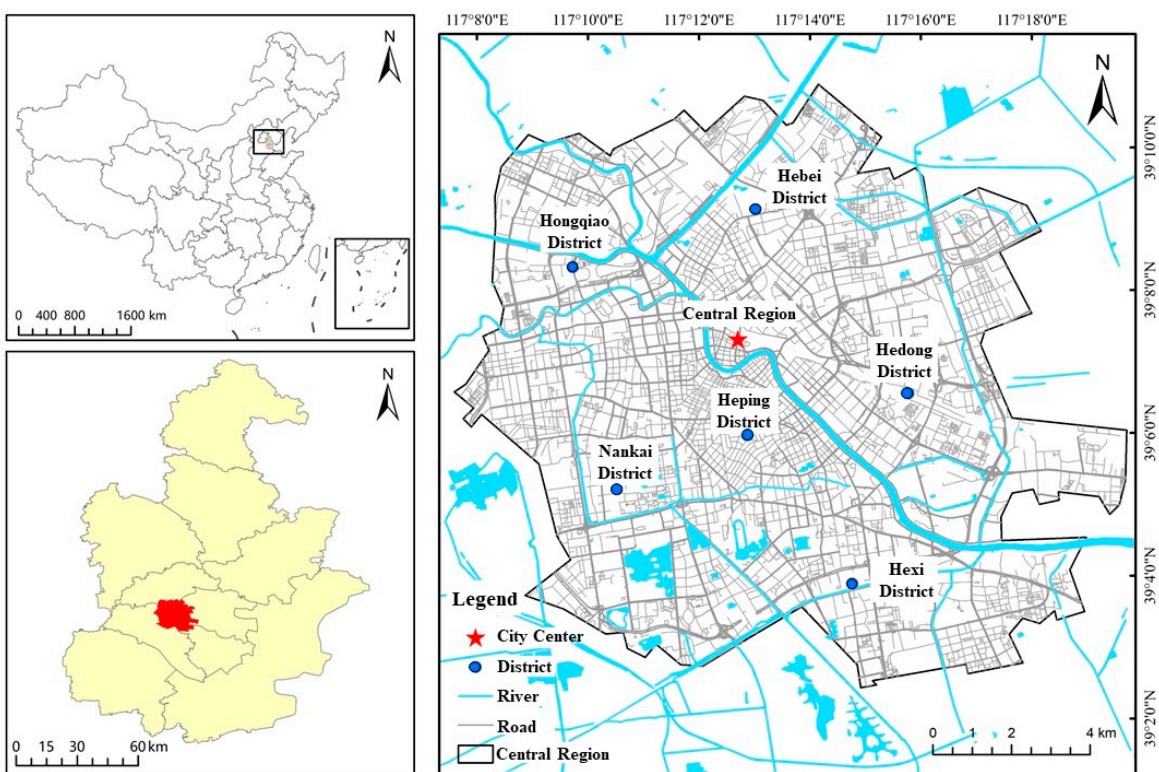

**Figure 1.** Study area.

### 2.2. Data Source

Tianjin's "pandemic period" began on 21 January 2020, with the confirmation of the first two positive cases [38]. Tianjin had experienced several pandemic waves since then. Following the discovery, control, gradual resumption of activity, and recovery phases, Tianjin faced China's first real battle against Omicron on 8 January 2022 [39]. After that, a "dynamic zero-COVID-19" strategy was implemented. Through the efforts of all parties, the State Council's joint prevention and control mechanism announced the Announcement on Further Optimizing and Implementing the Prevention and Control Measures of COVID-19 Pandemic on 7 December 2022, signaling a loosening of controls over the pandemic [40]. As a result, we divided the COVID-19 timeline into two phases: the pre-pandemic phase (before 8 January 2020), and the post-pandemic phase (from 7 December 2022 onwards). Hence, we obtained fundamental data from both the pre- and post-pandemic periods.

The study's foundational data were the geographical locations of parcel lockers and the social characteristics of the six central districts. We chose 19 December 2019 and 19 December 2022 as pre- and post-pandemic dates, respectively, and obtained geographical data for SPLs via electronic map, including name, address, longitude, and latitude. The social characteristics of the six central districts were derived from a variety of data sources, including urban development (UD, X1), built environment (BE, X2), and personal characteristics (PC, X3). Total population (X11), population density (X12), gender ratio (X31), population aging (X32), and per capita income (X33) were derived from the People's Republic of China's Population Census 2020 and Population Census 2022; road density (X21) was derived from OpenStreetMap (OSM) through this website (https://www.openstreetmap.org/ (accessed on 19 December 2022)), ArcGIS Data, and the Tianjin City Master Plan (2005–2020) [41], which was available to the general public on the government's official website; building density (X22) was derived from OpenStreetMap (OSM) and ArcGIS Data; residential community (X23) was derived from Lianjia.com, which was one of the largest real estate transaction service platforms in China (Table 1). Furthermore, correlation analysis in the EViews software package revealed that there was no multi-collinearity among these 8 variables (Table 2).

**Table 1.** Influencing factors, variables, and data sources of social characteristics of the 6 central districts in Tianjin.

| Factors | Variables | Data Sources |
|---|---|---|
| Urban development (UD, X1) | X11—Total Population | Population Census of Tianjin (2020, 2022) |
| | X12—Population Density | Population Census of Tianjin (2020, 2022) |
| Built environment (BE, X2) | X21—Road Density | OpenStreetMap (OSM), ArcGIS Data and Tianjin City Master Plan (2005–2020) |
| | X22—Building Density | OpenStreetMap (OSM) and ArcGIS Data |
| | X23—Residential Community | Lianjia.com (accessed on 19 December 2022) |
| Personal characteristics (PC, X3) | X31—Gender Ratio | Population Census of Tianjin (2020, 2022) |
| | X32—Population Aging | Population Census of Tianjin (2020, 2022) |
| | X33—Per capita income | Population Census of Tianjin (2020, 2022) |

**Table 2.** Correlations of variables in EViews.

| | X11 | X12 | X21 | X22 | X23 | X31 | X32 | X33 |
|---|---|---|---|---|---|---|---|---|
| X11 | 1.000 | | | | | | | |
| X12 | −0.645 | 1.000 | | | | | | |
| X21 | −0.420 | 0.793 | 1.000 | | | | | |
| X22 | −0.363 | 0.707 | 0.842 | 1.000 | | | | |
| X23 | 0.586 | 0.223 | 0.268 | 0.230 | 1.000 | | | |
| X31 | 0.141 | −0.786 | −0.692 | −0.544 | −0.639 | 1.000 | | |
| X32 | −0.295 | 0.536 | 0.534 | 0.490 | 0.118 | −0.643 | 1.000 | |
| X33 | −0.323 | 0.072 | 0.090 | 0.130 | −0.248 | −0.066 | 0.424 | 1.000 |

*2.3. Methods*

2.3.1. Kernel Density Estimation Method

The Kernel Density Estimation (KDE) method is a non-parametric method for estimating a random variable's probability density function. The basic principle of KDE is to compute a magnitude-per-unit area from point characteristics in order to generate a smoothly tapered surface. Density surfaces highlight areas where point characteristics are concentrated, making it easier to locate occurrence hotspots [42]. A growing body of literature demonstrates the utility of KDE in exploratory data analysis, statistical modeling, and machine learning for visualizing the underlying distribution of a dataset, assessing model fit, and generating synthetic data [43,44]. KDE is used in this study to determine the spatial agglomeration characteristics of SPLs distribution. The higher the kernel density

grade, the denser the point distribution and the lower the scattering. The equation is as follows [45]:

$$f(s) = \sum_{i}^{n} \frac{k}{\pi \gamma^2} \left( \frac{d_{is}}{\gamma} \right) \tag{1}$$

where $f(s)$ is the density at position $s$; $d_{is}$ is the distance from $i$ to position $s$; and $k$ is the weight of $d_{is}$. The search radius for the core density estimate, $r$, equals 500 m.

### 2.3.2. Average Nearest Neighbor Analysis

The spatial distribution randomness of SPLs in central Tianjin was determined using the quantitative technique of average nearest neighbor (ANN) analysis, which is a statistical measure used to evaluate the spatial distribution pattern of a set of points in two dimensions. The ANN index was calculated using Equation (2) [46,47]:

$$R = \frac{\overline{D_o}}{\overline{D_r}} \tag{2}$$

where $R$ denotes the ANN index, a geographical index that indicates the degree of mutual proximity in geographical space and can accurately reflect the spatial distribution characteristics of point-like elements. If $R = 1$, the SPLs are randomly distributed. If $R > 1$, the SPLs tend to be dispersed distributed, as points are farther apart than they would be if randomly distributed. If $R < 1$, the SPLs have a clustered distribution because the points are closer to each other than if they were randomly distributed [48]. Additionally, $D_o$ is the Observed Mean Distance (OMD), which is the average distance between each point and its nearest neighboring point; and $D_r$ is the Expected Mean Distance (EMD) for an identical number of points randomly distributed across the same space, and the equation is given by:

$$\overline{D_r} = \frac{1}{\sqrt{\frac{N}{A}}} \tag{3}$$

where $N$ denotes the number of research subjects, which refers to SPLs in this study; and $A$ denotes the geographical area.

### 2.3.3. Standard Deviational Ellipse

The standard deviational ellipse (SDE) method is a statistical method that is widely used to explore and analyze the spatial variation trends of geographic elements such as the gravity center, distribution, orientation, and shape [49,50]. The SDE method can intuitively describe the spatial and temporal evolution trends of geographic elements from a variety of perspectives, including ellipse center coordinates, rotation angle, and major and minor axe standard deviations [51]. In this study, the SDE method is used to identify the development trend of the spatial distribution of SPLs in the study area during pre- and post-pandemic periods. The Equation (4) is [52].

$$SDE_x = \sqrt{\frac{\sum_{i=1}^{n} (x_i - \overline{x})^2}{n}}, \ SDE_y = \sqrt{\frac{\sum_{i=1}^{n} (y_i - \overline{y})^2}{n}} \tag{4}$$

where $x_i$ and $y_i$ are the coordinates for feature $i$, $\overline{x}$ and $\overline{y}$ represent the mean center for the features, and $n$ equals to the total number of features. The ellipse's direction is determined by the $x$ axis; north is 0 degrees, clockwise rotation. The angle of rotation was calculated as Equations (5) and (6):

$$\tan \theta = \frac{A + B}{C} \tag{5}$$

$$A = \sum_{i=1}^{n} \tilde{x}_i^2 - \sum_{i=1}^{n} \tilde{y}_i^2, \ B = \sqrt{\left( \sum_{i=1}^{n} \tilde{x}_i^2 - \sum_{i=1}^{n} \tilde{y}_i^2 \right)^2 + 4 \left( \sum_{i=1}^{n} \tilde{x}_i \tilde{y}_i \right)^2}, \ C = 2 \sum_{i=1}^{n} \tilde{x}_i \tilde{y}_i \tag{6}$$

where $\widetilde{x}_i$ and $\widetilde{y}_i$ are the x- and y-coordinate deviations from the mean center, respectively.

### 2.3.4. Geographical Detector

The Geographical Detector (GD) was proposed by Wang et al. [53] to identify the driving force by detecting spatial heterogeneity [50,54]. The central idea is based on the assumption that if an independent variable (X) has a significant influence on a dependent variable (Y), their spatial distributions should be similar. The GD method has been widely used to quantitatively analyze the driving factors of risk region spatial patterns [55–57], environmental changes [58,59], and identify temporal–spatial differences [60,61]. The four detectors in GD are as follows: differentiation and factor detector, risk detector, interaction detector, and ecological detector. In this study, factor detectors and interaction detectors are used.

The factor detector reveals the relative importance of explanatory variables with the q-statistic, which compares the dispersion variances between observations in the entire study area and strata of variables. The *q*-value of a potential variable is calculated as follows [62]:

$$q = 1 - \frac{1}{N\sigma^2} \sum_{h=1}^{L} N_h \sigma_h^2 \qquad (7)$$

where *h* and *L* are the stratification of variable Y or factor X; the $N_h$ and $N$ are the number of units in the stratification h and the whole area, respectively; $\sigma_h^2$ and $\sigma^2$ are the variance of the Y values of the h stratification and the total area, respectively; and *q* is the force of the influential factors on SPLs spatial distribution. The range of *q* values is [0, 1], and the closer the *q*-value is to 1, the stronger the explanatory power of this influencing variable X to the attribute Y, and vice versa.

The interaction detector determines the interactive impacts of two overlapped spatial variables based on the relative importance of interactions computed with *q*-values of the factor detector [53,63]. When the results are combined, they can determine whether the two variables weaken or strengthen each other, or whether they are mutually independent in causing desertification. Specific types of interaction are shown in Table 3 [54,57,64]. The results of interaction detectors could compensate for the shortcoming of measuring the effect of a single factor on dependent variables while ignoring the interaction between multiple factors.

**Table 3.** Interaction types between two variables.

| GD Results | Type of Interaction |
|---|---|
| q(X1 ∩ X2) < Min(q(X1), q(X2)) | Nonlinear-weaken |
| Min(q(X1), q(X2)) < q(X1 ∩ X2) < Max(q(X1), q(X2)) [1] | Uni-variable weaken |
| q(X1 ∩ X2) > Max(q(X1), q(X2)) | Bi-variable enhance |
| q(X1 ∩ X2) = q(X1) + q(X2) | Independent |
| q(X1 ∩ X2) > q(X1) + q(X2) | Nonlinear-enhance |

[1] Min(q(X1), q(X2)) means the minimum of q(X1) and q(X2); Max(q(X1), q(X2)) indicates the maximum of q(X1) and q(X2).

## 3. Results

### 3.1. Differences in the Quantitative Characteristics between Pre- and Post-Pandemic

The number of SPLs increased dramatically between the pre- and post-pandemic periods (Figure 2). The number of SPLs in the study area increased from 51 to 479 between the end of 2019 and the end of 2022, with an average annual growth rate of about 279.74%. It should be noted that, while the total number of facilities has increased, we carefully checked and discovered that some SPLs that existed in 2019 did not exist in 2022. Out of the total 51 SPLs in 2019, only 13 SPLs continued to operate in 2022. This indicates that 38 SPLs (75% of the SPLs in 2019) were discontinued within three years, with two of them located in office buildings and the rest in residential areas.

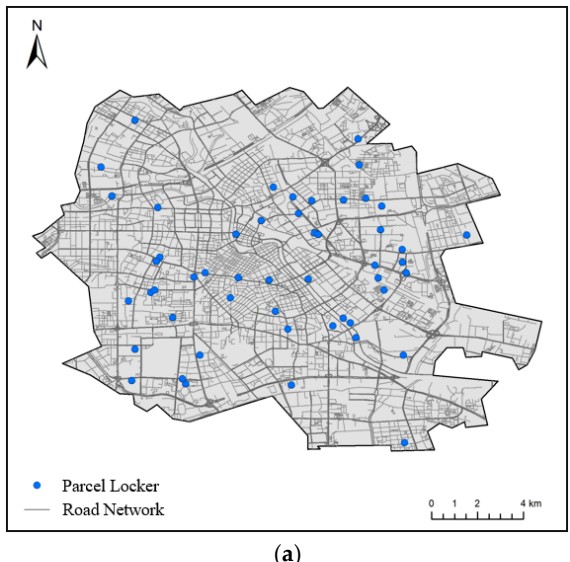 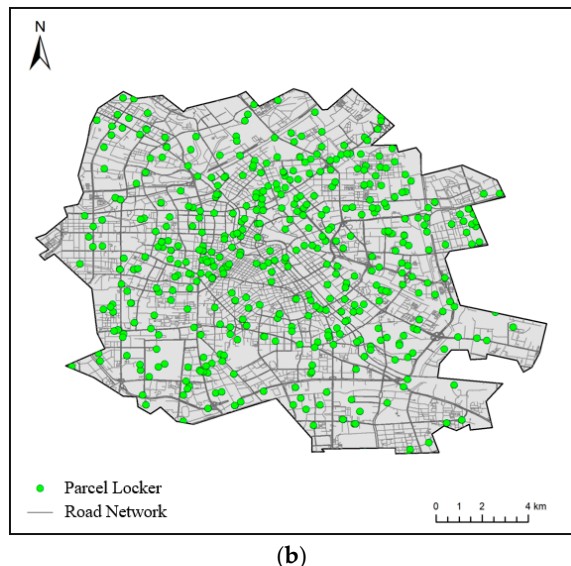

**Figure 2.** Spatial Distribution of SPLs in the central Tianjin region: (**a**) SPLs spatial distribution in 2019; (**b**) SPLs spatial distribution in 2022.

With regard to the clawed data of address, we classified SPLs application scenario in Tianjin into five categories: residential community, office building, shopping mall, school, and historical and cultural block (Figure 3). In 2019, PLs were used in three categories: residential communities (47, 92.2%), office buildings (3, 5.9%), and shopping malls (1, 1.9%). The difference between 2022 and 2019 was not only in the increasing number of SPLs, but also in the types of SPLs. In 2022, the residential communities still had the most SPLs (468, 97.7%), followed by the office buildings (5, 1.0%) and the shopping malls (4, 0.8%). There are two new categories: school (1, 0.25%) and historical and cultural block (1, 0.25%).

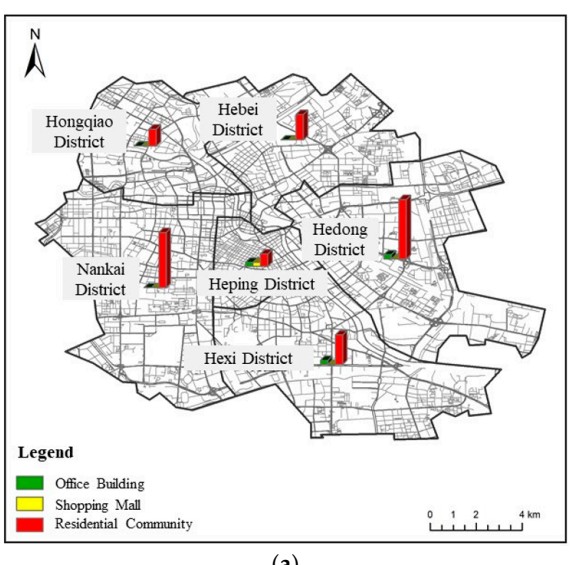 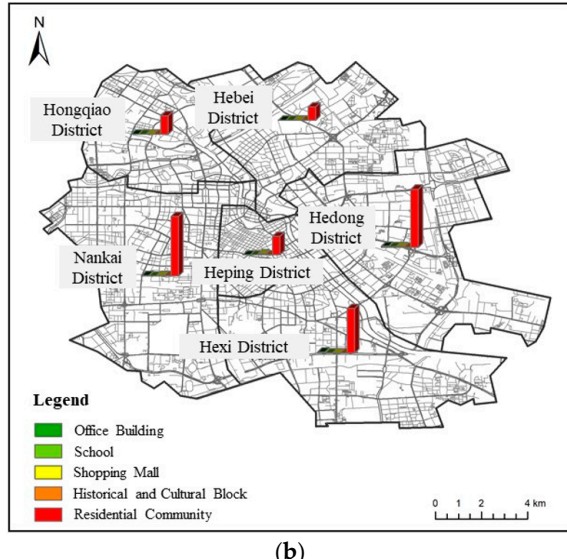

**Figure 3.** SPLs application scenario in the central Tianjin region: (**a**) SPLs application scenario in 2019; (**b**) SPLs application scenario in 2022.

To further investigate the spatial distribution level from a per capita perspective, we calculated the per capita SPL possession amounts through dividing the total number of SPLs by the total registered population at the end of the year. Between 2019 and 2022, there was a nearly tenfold increase: every 80,280 people in the central Tianjin region had one SPL in 2019, while every 8793 people had one in 2022.

### 3.2. Differences in the Spatial Agglomeration Characteristics between Pre- and Post-Pandemic

In order to further illuminate the spatial agglomeration characteristics of SPLs in the central Tianjin region, we calculated and visually expressed the KDE of 2019 and 2022 (Figure 4).

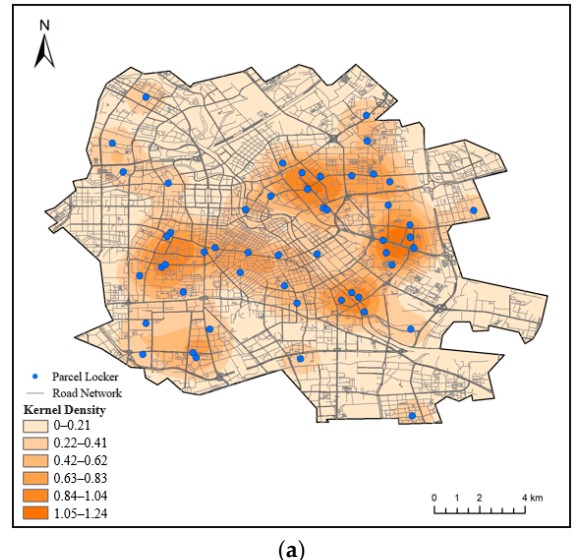

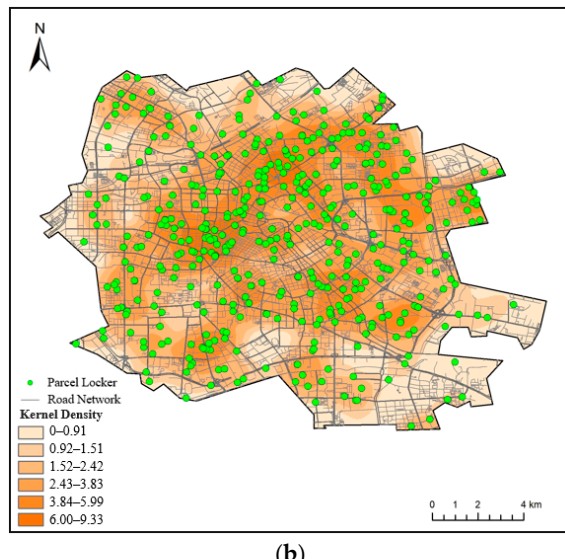

**Figure 4.** KDE of SPLs in the central Tianjin region: (**a**) KDE in 2019; (**b**) KDE in 2022.

The results of density analysis shows that the SPLs in the central Tianjin region has formed a circular pattern of "core circle-edge belt". In 2019, an obvious annular high-value area appeared, which perfectly corresponds to Tianjin's outer ring road. Due to the annular area, there were two low-density areas: one surrounded by the center, which primarily involved the Heping District, and the other was the "edge belt", which included Hongqiao District and the border areas between other districts and Tianjin's suburbs. By the end of 2022, the annular "core circle" had gradually expanded its bandwidth and covered a larger area. Concurrently, the density value of the enclosed area of the core circle has clearly increased, forming a structure with the "core circle" in which the density value gradually increases from inside to outside. The border area between the central city and the suburbs, however, remains low-density.

### 3.3. Differences in the Spatial Equilibrium Characteristics between Pre- and Post-Pandemic

Figure 5 depicts the results of the ANN analysis. In 2019, the ANN index was 1.03, the *p*-value was 0.65, and the z-score was less than one, indicating that the SPLs distribution was obviously random. In 2022, the ANN index was 0.82, the *p*-value was 0, and the z-score was −7.28, indicating that the SPLs distribution had clear clustering characteristics.

The SDE was carried out using ArcGIS to further investigate the evolving pattern of SPLs from 2019 to 2022 (Figure 6). As the analysis results of SPLs data in 2019 show, an ellipse with a center at 39°7′15″ N and 117°12′18″ E (Hezuoli Residential Community, Heping District, Tianjin), left-handed at 76.08° counterclockwise, 4724.07 m as the major axis and 4228.89 m as the minor axis, was shown in the figure by the blue line. The green line represents the SPL analysis results in 2022. The ellipse had a center at 39°7′35″ N and 117°12′19″ E (Huayin Nanli Residential Community, Heping District, Tianjin), a circumference of 133.07° counterclockwise, a major axis of 4934.15 m, and a minor axis of 4724.73 m. To summarize, the ellipse's center clearly deviates to the north; the distribution of SPLs gradually exhibits a "north-west-southeast" trend, which is consistent with the direction Haihe River flows and also with Tianjin's central city development pattern.

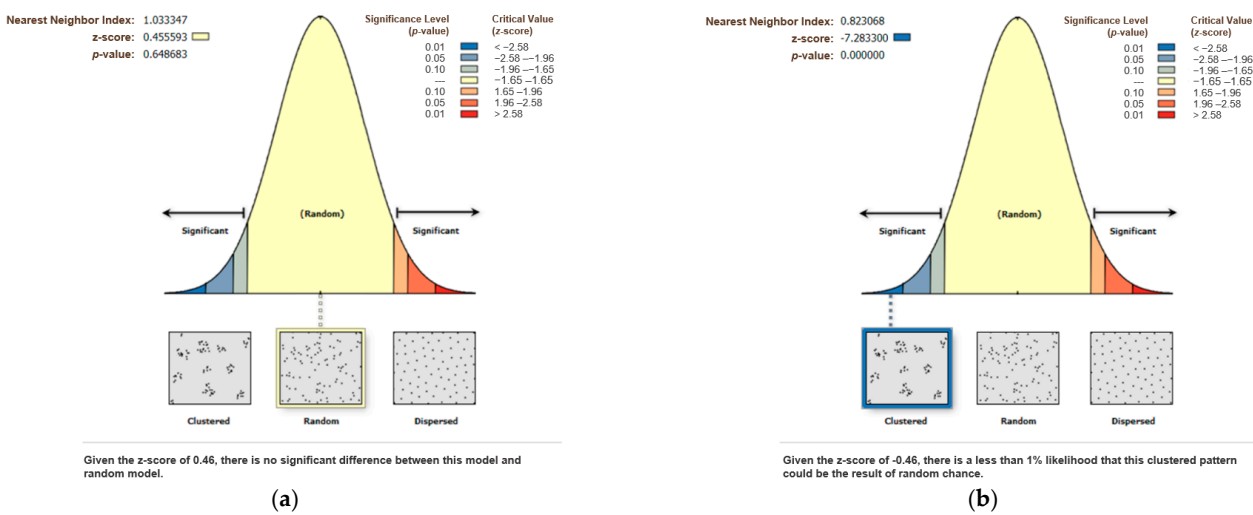

**Figure 5.** ANN analysis of SPLs in Tianjin's Central Urban: (**a**) ANN in 2019; (**b**) ANN in 2022.

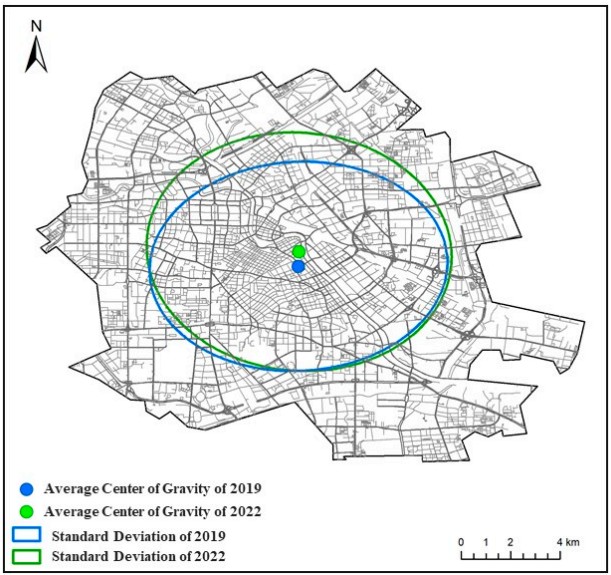

**Figure 6.** SDE of SPLs in the central Tianjin region.

### 3.4. Differences in the Influencing Factors between Pre- and Post-Pandemic

3.4.1. Factor Detection

According to the results of the geographical detector calculation (Table 4), all *p*-values were significant at the 5% and 1% levels. In 2019, the top three variables with relatively high associations with the development of SPLs spatial distribution were per capita income (0.894), total population (0.738), and population aging (0.515). In 2022, the top three variables were total population (0.838), per capita income (0.725), and building density (0.651).

In both 2019 and 2022, the total population had a much higher influence intensity than population density in the category of urban development. The order of influence intensity in the category of built environment was X22 > X21 > X23 in both 2019 and 2022, indicating that building density was a key influential factor. In the category of personal characteristics, the order of influence intensity was X33 > X32 > X31, indicating that per capita income was a key influential factor. It is worth noting, however, that the influence intensity of the gender ratio increased to be close to the population aging.

**Table 4.** The q-statistic and *p*-value of influential factors in 2019 and 2022.

| | | Year of 2019 | | Year of 2022 | |
|---|---|---|---|---|---|
| | | q-Statistic [1] | *p*-Value | q-Statistic | *p*-Value |
| X1_UD | X11_TP | 0.738 *** | 0.0000 | 0.838 *** | 0.0000 |
| | X12_PD | 0.200 ** | 0.0478 | 0.421 *** | 0.0000 |
| X2_BE | X21_RD | 0.322 ** | 0.0497 | 0.497 ** | 0.0026 |
| | X22_BD | 0.364 *** | 0.0035 | 0.651 *** | 0.0000 |
| | X23_RC | 0.279 ** | 0.0119 | 0.398 *** | 0.0000 |
| X3_PC | X31_GR | 0.200 ** | 0.0478 | 0.444 *** | 0.0000 |
| | X32_PA | 0.515 *** | 0.0000 | 0.464 ** | 0.0046 |
| | X33_PI | 0.894 *** | 0.0000 | 0.725 *** | 0.0000 |

[1] *** significant at the 1% level; ** significant at the 5% level.

### 3.4.2. Interaction Detection

The influence of the bi-factors could not be ignored because the variables are not independent of one another. The Geographical Detector results (Table 5) show that the influence of driving factors on SPLs distribution change was realized through the interaction of multiple factors rather than a single factor, because the results of bi-variable interaction were bi-variable enhanced and nonlinear enhanced in both 2019 and 2022, and the influence of bi-variable interaction was stronger than that of a single factor.

**Table 5.** GD interaction detection results in 2019 and 2022.

| Bi-Factors | Year of 2019 | | Year of 2022 | |
|---|---|---|---|---|
| | Result | Interaction Type [1] | Result | Interaction Type |
| X11_TP ∩ X12_PD | 0.880 | BE | 1 | BE |
| X11_TP ∩ X21_RD | 0.765 | BE | 0.909 | BE |
| X11_TP ∩ X22_BD | 0.880 | BE | 0.909 | BE |
| X11_TP ∩ X23_RC | 0.880 | BE | 0.843 | BE |
| X11_TP ∩ X31_GD | 0.880 | BE | 0.843 | BE |
| X11_TP ∩ X32_PA | 0.995 | BE | 0.933 | BE |
| X11_TP ∩ X33_PI | 0.976 | BE | 0.999 | BE |
| X12_PD ∩ X21_RD | 0.535 | BE | 0.797 | BE |
| X12_PD ∩ X22_BD | 0.880 | NE | 0.893 | BE |
| X12_PD ∩ X23_RC | 0.492 | BE | 1 | NE |
| X12_PD ∩ X31_GD | 0.492 | NE | 0.850 | BE |
| X12_PD ∩ X32_PA | 0.535 | BE | 0.491 | BE |
| X12_PD ∩ X33_PI | 0.923 | BE | 1 | BE |
| X21_RD ∩ X22_BD | 0.980 | NE | 0.702 | BE |
| X21_RD ∩ X23_RC | 0.612 | BE | 0.909 | BE |
| X21_RD ∩ X31_GD | 0.342 | BE | 0.526 | BE |
| X21_RD ∩ X32_PA | 0.535 | BE | 0.759 | BE |
| X21_RD ∩ X33_PI | 0.923 | BE | 0.793 | BE |
| X22_BD ∩ X23_RC | 0.880 | NE | 0.909 | BE |
| X22_BD ∩ X31_GD | 0.861 | NE | 0.909 | BE |
| X22_BD ∩ X32_PA | 1 | NE | 0.686 | BE |
| X22_BD ∩ X33_PI | 1 | BE | 0.792 | BE |
| X23_RC ∩ X31_GD | 0.492 | BE | 0.843 | BE |
| X23_RC ∩ X32_PA | 0.568 | BE | 0.933 | NE |
| X23_RC ∩ X33_PI | 0.956 | BE | 0.808 | BE |
| X31_GD ∩ X32_PA | 0.612 | BE | 0.784 | BE |
| X31_GD ∩ X33_PI | 1 | BE | 1 | BE |
| X32_PA ∩ X33_PI | 0.916 | BE | 0.793 | BE |

[1] BE stands for bi-variable Enhancement; and NE stands for nonlinear enhancement.

In 2019, there were ten bi-factor groups with interaction coefficients greater than 0.9: X11 ∩ X32, X11 ∩ X33, X12 ∩ X33, X21 ∩ X22, X21 ∩ X33, X22 ∩ X32, X22 ∩ X33, X23 ∩ X33,

X31 ∩ X33, and X32 ∩ X33, which had more significant effects on the spatial distribution of SPLs than other bi-factor groups. Among these, X22 ∩ X32, X22 ∩ X33, and X31 ∩ X33 had the most sway, indicating that the interaction of per capita income, building density, gender ratio, and population aging had the greatest explanatory power on the spatial distribution of SPLs in the central Tianjin region.

In comparison to the results of interaction detection in 2019, there were 12 interaction groups with results greater than 0.9 in 2022: X11 ∩ X12, X11 ∩ X21, X11 ∩ X22, X11 ∩ X32, X11 ∩ X33, X12 ∩ X23, X12 ∩ X33, X21 ∩ X23, X22 ∩ X23, X22 ∩ X31, X23 ∩ X32, X31 ∩ X33. Among these, X11 ∩ X12, X12 ∩ X23, X12 ∩ X33, and X31 ∩ X33 had the most influence, indicating that the interaction of population density, total population, residential community, per capita income, and gender ratio had the most explanatory power on the spatial distribution of SPLs in the study area.

A comparison of heat maps also revealed a significant shift in the factors influencing the spatial distribution of SPLs (Figure 7). In 2019, the q-value zones for the factors interacting with X33 per capita income, X11 total population, and X22 building density were obviously high, indicating that city logistics development was consistent with people and capital flow. The q-value zone for X11 total population remained high in 2022, while X33 per capita income and X22 building density decreased. Furthermore, in 2022, the q-value for the X23 residential community was increased, confirming the previously stated views that SPLs were most commonly used in residential communities.

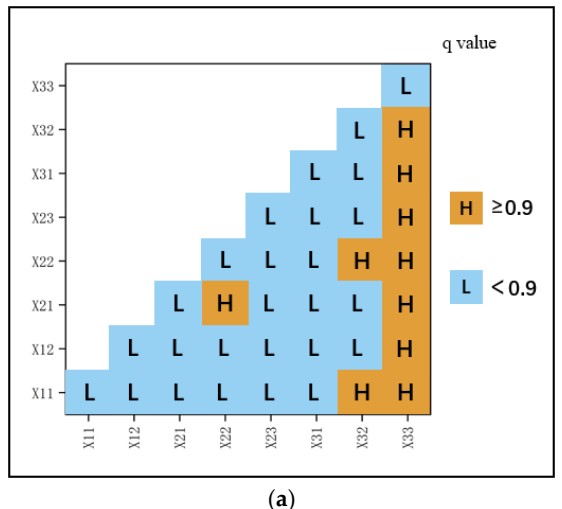

(a)

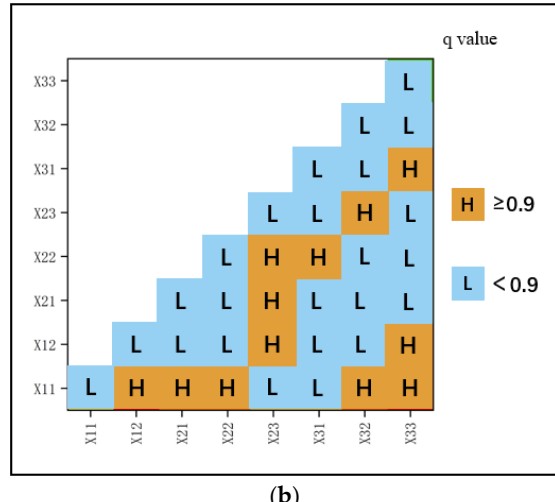

(b)

**Figure 7.** Interaction detection heat map: (**a**) heat map in 2019; (**b**) heat map in 2022.

## 4. Discussion

Prior to the pandemic, the traditional retail industry was already being impacted by the rise of online shopping. If it had not been for COVID-19, the introduction of SPLs would have been much slower. In central Tianjin, for example, just five SPLs were created as demonstration projects in residential areas in 2016. However, by 2019, the number had risen to 51, and by 2022, the area had 479 SPLs. The unexpected pandemic breakout has expedited the expansion of e-commerce and caused a shift in the way many people live, work, and shop, encouraging the rise of self-service delivery options such as SPLs [65,66]. As such, we chose SPL as one of the representatives of the SSFs, analyzed the temporal and spatial differences before and after the pandemic, and identified the factors that influence the spread of SPLs. This section will discuss SPLs application scenarios, layout planning issues, and current SSF research trends that may be able to solve these problems.

### 4.1. SPLs Application Scenearios

The e-commerce industry has been rapidly expanding for many years. The COVID-19 pandemic accelerated this trend, exacerbating the market for courier and express services [1].

As a low-cost, self-service, and low-carbon last-mile logistics solution, SPL has become popular around the world [9,21,28–30]. During the pandemic, the e-commerce in Pakistan increased by 10% in daily record [1]; the volume of parcels increased by 37% in the UK from April to May 2020 [67]; the number of parcels delivered in Poland increased by 20–100%, and the number of SPLs increased from 7000 to 13,000 [18]. Moreover, the application scenario for SPLs can range from residential buildings to office buildings, train stations, and shopping malls [68].

Many studies on the location and allocation of SPLs have been conducted in order to improve service and lower the cost of last-mile delivery. None, however, have reached a conclusion on the most concentrated application scenario of SPLs. In this study, we discovered a significant difference between pre- and post-pandemic conditions. From 2019 to 2022, the number of SPLs in central Tianjin increased dramatically from 51 to 479, implying that the per capita of SPLs increased from 80,280 to 8793 people/SPL. The majority of SPLs in central Tianjin were in residential areas (92.2% in 2019, and 97.7% in 2022). Our findings revealed that the majority of SPLs were installed in residential communities, which could support one of the most frequently stated preferences of consumers for SPL locations: proximity to their home [69].

Moreover, in 2022, one SPL was located in a school and one SPL was located in the historical and cultural block, according to this study. School is a place where teachers and students spend a lot of time during the day, and it could be considered one of the last-mile logistics nodes to end-consumers. Similarly, the one SPL in the historical and cultural block could provide temporary storage for nearby residents, store operators, and visitors. SPLs are used for parcel delivery and collection. Although the SPLs are primarily designed to function as collection and delivery points, they have gradually evolved to serve a variety of other purposes, such as temporary storage, online payment, and even as cash machines. These potential functional transformations of SPLs can be seen as potential profit methods and should also be taken into consideration when planning for their location.

### 4.2. SPLs Layout Planning Issues

SPLs spatial distribution changed dramatically before and after the pandemic, and the factors influencing the SPLs distribution are multilevel and their effects are complex [12]. Before the pandemic, the spatial distribution of SPLs in the central Tianjin region was random, with per capita income, building density, gender ratio, and population aging having the most explanatory power on the spatial distribution. During the pandemic, the gravity center of the SPLs clusters shifted from south to north, with a "north-west-southeast" development trend that corresponded to the Haihe River and Tianjin's urban development patterns. After the pandemic, the KDE and ANN results revealed that the spatial distribution of SPLs had changed from random to agglomerated, with the factors of population density, total population, residential community, per capita income, and gender ratio having the greatest explanatory power on the spatial distribution of SPLs in the study area.

These transformations occurred in just three years. To put it another way, the spatial distribution of SPL is highly uncertain. Here, we are going to perform a calculation: assuming that SPLs have a uniform standard size and that each SPL is the same size, and using the SPL size adopted by [70], the schematic of a parcel locker is 61 cm (W) × 366 cm (L) × 183 cm (H), and at least 3 m must be reserved in front of an SPL to make it convenient for customers to pick up their parcels. Then, the minimum amount of space needed for an SPL is 361 cm (W) × 366 cm (L) × 183 cm (H). That is, over the course of three years, SPLs occupied at least 5655 m$^2$ of residential space. In fact, because of the different population density and demands of each residential area, the size and space required by SPLs are frequently larger, which is incredible in a high-density mega-city's central region. It should also be noted that this does not include the parking space needed for the courier's delivery. This phenomenon could even be called "the invasion of public space by SPLs". Since the decision on the optimal number, locations, and sizes of SPLs has a significant impact on

public spaces, urban planning intervention should be improved to avoid missing or lagging behind in this process [29].

According to studies [3,4], for many, an abrupt disaster could lead to dramatic changes in attitudes and behavior, which compose a new normal of lifestyle; for some, life can return to old normal; for others, however, their life can be still in transition for a long time. Furthermore, Lagorio, A. and Pinto, R. [71] identified availability, accessibility, security, environmental impact/land occupation, costs, methods of use, and regulations as influential factors and issues in SPLs location. Under these uncertain conditions, how should we plan the spatial layout of SPL? What kind of space should we choose to place SPL if the demand for it grows? What should be done with abandoned SPL space if the demand for SPL falls?

*4.3. SPLs Dynamic Layout Planning*

There is no doubt that, regardless of future trends, continued large-scale SPL expansion will result in supply exceeding demand. As previously stated, both customer demand and the environment are uncertain. In this case, tracking dynamic changes in consumer demand and developing dynamic layouts of SSFs such as SPLs will be a very practical trend.

According to current SPL layout research trends, dynamic planning could be an effective method to deal with uncertain conditions. There are two ways to realize dynamic planning: dynamic planning and mobile facilities [72,73]. Dynamic planning, as opposed to traditional static planning, divides the planning process into several short-term stages, forecasts demand for each stage, and makes corresponding planning adjustments [74], which is suitable for uncertain situations. To realize dynamic planning, heuristic and meta-heuristic algorithms are commonly used to model and calculate basic data. Another approach is to use mobile facilities, which entails converting fixed SPLs to mobile parcel lockers, autonomous delivery robots, and delivery drones that can change locations throughout the day to increase customer accessibility [10,11,75–78]. If all stages and delivery routes are carefully planned, dynamic planning could be an effective solution to address the encroachment of public spaces caused by SPLs.

**5. Conclusions**

In the shock of the public health emergency, the use of public space underwent a subversive transformation. An epidemic always has a time limit, but its impact on human lifestyle is far-reaching. In the long run, this will mark the beginning of a lifestyle shift: residential communities will become the focal point of people's lives; the online scenario will become the primary working and shopping scenario; and simplification and self-service are necessary to all ages. That is why the SPLs, representative of SSFs, are increasingly widely applied in cities. To address the changes between pre- and post-pandemic periods, this study investigated the spatial distribution characteristics of SPLs and their influential factors based on the geographical data of SPLs and social characteristics data of the central Tianjin region. The findings will contribute to a better understanding of the evolution of SSFs. This study yielded the following conclusions.

(1) The COVID-19 pandemic has accelerated the emergence of SPLs, and the majority of SPLs clustered in residential communities.

(2) The spatial distribution of SPLs experienced the process from random to agglomerated during the pre- and post-pandemic period, and the development trend of SPLs coincided with Tianjin's urban development patterns.

(3) During the three-year course, the total population remained one of the most influential factors, per capita income and building density became less influential, and most importantly, the influential power of residential community increased, which deserves more attention. The evolution of influencing factors demonstrates that the emphasis on SPLs layout has shifted from cost reduction and efficiency improvement to customer-orientation.

However, this study also has several limitations, which should be addressed in future studies. To begin with, the study only obtained data for 2019 and 2022, which may result in a gap between 2019 and 2022 and the inability to show the entire process of SPL development. In this regard, future research could broaden the timeline of SPLs' evolution and refine the development of each stage. This study's scope is limited to SPLs in central Tianjin region. Future researchers could conduct community-level research to gain a micro perspective or conduct a macro perspective comparative analysis with other cities.

**Author Contributions:** Conceptualization, Mengyue Ding and Yike Hu; Methodology, Mengyue Ding; Software, Mengyue Ding and Nadeem Ullah; Formal analysis, Mengyue Ding, Nadeem Ullah and Sara Grigoryan; Investigation, Yan Song; Resources, Mengyue Ding; Data curation, Mengyue Ding, Nadeem Ullah and Sara Grigoryan; Writing—original draft, Mengyue Ding; Writing—review & editing, Mengyue Ding, Nadeem Ullah, Sara Grigoryan, Yike Hu and Yan Song; Visualization, Mengyue Ding, Nadeem Ullah and Sara Grigoryan; Supervision, Yike Hu; Project administration, Yike Hu and Yan Song; Funding acquisition, Yike Hu and Yan Song. All authors have read and agreed to the published version of the manuscript.

**Funding:** This research was funded by Reconstructing the Architecture System based on the Coherence Mechanism of "Architecture-human-environment" in the Chinese Context, Key Project of National Natural Science Foundation of China, grant number 52038007, and "Science and Technology R&D Platform Construction Special Project of Hebei Province, China (21567631H)".

**Institutional Review Board Statement:** Not applicable.

**Informed Consent Statement:** Not applicable.

**Data Availability Statement:** The data used to support the findings of this study are available from the authors upon request.

**Acknowledgments:** The authors wish to express their appreciation and gratitude to the anonymous reviewers and editors for their insightful comments and suggestions to improve the paper's quality.

**Conflicts of Interest:** The authors declare no conflict of interest.

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
