# Peer review of "Variations in the Spatial Distribution of Smart Parcel Lockers in the Central Metropolitan Region of Tianjin, China: A Comparative Analysis before and after COVID-19"

_ijgi, doi:10.3390/ijgi12050203_

Round 1
Reviewer 1 Report
Thanks for your very interesting manuscript. Only a few comments from me, with the aim of improving it :
· Page 2, Introduction, you write: "Despite the fact that many studies on SPLs have been published, to the best of the authors’ knowledge, no study has attempted to analyze the temporal and spatial differences before and after the pandemic to depict the spread of SPLs. These are the major gaps in knowledge that this study will try to fill."
I think here, you can add that you are looking for the factors that influence these changes.
· Page 7, Results 3.1, you write about the average annual growth rate. Two questions, first, do you have the annual growth rate before and during the pandemic? And second, about the SPLs that did not exist in 2022, were in residential communities, office buildings or shopping malls?
· Page 11, Results 3.4.2, you write that gender ratio had most explanatory power on the spartial distribution of SPLs. Is this due to the correlation between the residential (community – office – mall) area type and gender ratio?
· Page 12. par. 4.1 you write: "Nevertheless, there is a diverse demand for SPL". Please explain a little more what you mean.
Reviewer 2 Report
The overall content of the manuscript is fine, the aim, method, discussion and conclusion of study are clear, however, there are slight area of improvement, as follows:
1. I think you should use "Smart Parcel Locker” in the title.
2. To add issue (the environmental or cost or lead time or ?) of SPL that suit with research aim in the abstract – why there is a need to study the before and after Covid?
Reviewer 3 Report
as a empirical research, the idea is good, you want to investigate how the spatial distribution of parcel lockers before and after covid
I have some suggestions:
for your 3.3 and 3.4 it is a little expected to illustrate in a more illustrative way. Your 3.1 and 3.2 is good and it is easy for readers to catch the gist. While for 3.3 and 3.4 there might be some spaces to make them more illustrative so that even an outsider can get your point
you mentioned in your title, it is a study before and after covid. but it seems there are not too much about the affect from covid. If there is no covid, will the developing pattern be the same?
